# Antimicrobial Drug-Resistant Gram-Negative Saprophytic Bacteria Isolated from Ambient, Near-Shore Sediments of an Urbanized Estuary: Absence of β-Lactamase Drug-Resistance Genes

**DOI:** 10.3390/antibiotics9070400

**Published:** 2020-07-10

**Authors:** Charles F. Moritz, Robert E. Snyder, Lee W. Riley, Devin W. Immke, Ben K. Greenfield

**Affiliations:** 1School of Public Health, University of California, Berkeley, CA 94720, USA; cfmoritz@gmail.com (C.F.M.); robert.snyder@berkeley.edu (R.E.S.); lwriley@berkeley.edu (L.W.R.); 2Department of Environmental Sciences, Southern Illinois University, Edwardsville, IL 62026, USA; dimmke@siue.edu

**Keywords:** antibiotic resistance, aquatic contamination, probabilistic sampling, San Francisco Estuary, coast, *Pseudomonas*, *Shewanella algae*, *Vibrio parahaemolyticus*

## Abstract

We assessed the prevalence of antimicrobial resistance and screened for clinically relevant β-lactamase resistance determinants in Gram-negative bacteria from a large urbanized estuary. In contrast to the broad literature documenting potentially hazardous resistance determinants near wastewater treatment discharge points and other local sources of aquatic pollution, we employed a probabilistic survey design to examine ambient, near-shore sediments. We plated environmental samples from 40 intertidal and shallow subtidal areas around San Francisco Bay (California, USA) on drug-supplemented MacConkey agar, and we tested isolates for antimicrobial resistance and presence of clinically relevant β-lactamase resistance determinants. Of the 74 isolates identified, the most frequently recovered taxa were *Vibrio* spp. (40%), *Shewanella* spp. (36%), *Pseudomonas* spp. (11%), and *Aeromonas* spp. (4%). Of the 55 isolates tested for antimicrobial resistance, the *Vibrio* spp. showed the most notable resistance profiles. Most (96%) were resistant to ampicillin, and two isolates showed multidrug-resistant phenotypes: *V. alginolyticus* (cefotaxime, ampicillin, gentamicin, cefoxitin) and *V. fluvialis* (cefotaxime, ampicillin, cefoxitin). Targeted testing for class 1 integrons and presence of β-lactam-resistance gene variants TEM, SHV, OXA, CTX-M, and *Klebsiella pneumonia* carbapenemase (KPC) did not reveal any isolates harboring these resistance determinants. Thus, while drug-resistant, Gram-negative bacteria were recovered from ambient sediments, neither clinically relevant strains nor mobile β-lactam resistance determinants were found. This suggests that Gram-negative bacteria in this well-managed, urbanized estuary are unlikely to constitute a major human exposure hazard at this time.

## 1. Introduction

The development of antimicrobial resistance in Gram-negative bacteria is a serious and growing global concern. Anthropogenic selection of highly resistant bacteria is driven by the overuse of antimicrobial agents in healthcare and agriculture as well as their mismanagement during waste disposal [1,2,3,4,5,6]. This selective process has dramatically affected global health; drug-resistant infections have become widespread globally [7,8,9,10] and were recently estimated at over 2 million infections in the United States annually [11]. Environmental and saprophytic bacteria are important as indicators and reservoirs of antibiotic resistance determinants that may be shared by human bacterial pathogens [12,13,14,15,16,17,18,19].

The β-lactams are currently the most widely used class of antimicrobial agents for treatment of bacterial infections in humans [20]. Gram-negative bacteria (GNB) have evolved to develop resistance to β-lactams by producing β-lactamase enzymes that hydrolyze β-lactams. Indeed, 2771 unique β-lactamase enzymes were discovered as of 2018 [21]. Extended-spectrum β-lactamases (ESBLs) such as TEM-, SHV-, OXA-, and CTX-M-type β-lactamases have become widespread in clinical and environmental settings, threatening the utility of broader-spectrum β-lactam drugs [21,22]. More recently, resistance to carbapenem drugs in GNB of the family Enterobacteriaceae, through production of *Klebsiella pneumonia* carbapenemases (KPCs), has become an imminent public health threat [9,14,23,24]. Genes encoding these β-lactamases are often located on mobile genetic elements that mediate their transfer between bacteria of the same or different species. This mechanism may contribute to dissemination of resistance determinants from the natural environment to healthcare settings [25].

Coastal and river waters located in populated areas with limited or overextended water and sanitation infrastructure harbor high rates of drug-resistant bacteria [6,17,26,27], but the extent to which this is true in areas with reliable secondary and tertiary wastewater treatment facilities is not as well characterized. San Francisco Bay (CA, USA) is located in a highly populated and urbanized region with extensive wastewater treatment infrastructure [28,29]. San Francisco Bay also has a legacy of environmental contamination that has resulted in elevated concentrations of a broad range of pollutants [30]. This includes fecal contamination observed at ponds managed as bird habitats and sloughs [31], and occasionally at swimming beaches [32,33]. Here, we assessed the prevalence of antimicrobial resistance in GNB in near-shore sediments collected from San Francisco Bay, an estuarine environment with ambient urban pollution. We determined resistant strain taxa and tested resistant isolates for class 1 integrons and presence of β-lactam-resistance gene variants TEM, SHV, OXA, CTX-M, and KPC. To our knowledge, this is the first probabilistic spatial survey of an estuary’s sediment for clinically relevant genetic resistance elements in GNB.

## 2. Results

None of the 40 collection sites were immediately adjacent to treated wastewater discharge locations (Figure 1). Bacteria colonies grew on unsupplemented plates from all 40 sites, but none presented as positive for lactose utilization, indicating that lac+ colonies (e.g., *Escherichia coli*) were absent from all samples. From the 40 sites, bacterial isolates that grew in the presence of ampicillin, gentamicin, imipenem, and cefotaxime were found at 34 (85%), 27 (67.5%), 15 (37.5%), and 9 (22.5%) sites, respectively (Table 1). From the initial antibiotic-containing MacConkey agar plates, 174 isolates were obtained and subjected to further analyses. Bacteria isolated from plates containing ampicillin were the most prevalent (87 isolates from 32 sites), followed by gentamicin (39 isolates from 13 sites), imipenem (37 from 15 sites), and cefotaxime (11 isolates from 8 sites) (Table 1).

Seventy-two different Gram-negative bacterial isolates were identified by their 16S rRNA sequences. They included 1 *Acinetobacter* sp. (1.4%), 3 *Aeromonas* spp. (4.2%), 1 *Castellaniella* sp. (1.4%), 1 *Gallaecimonas* sp. (1.4%), 8 *Pseudomonas* spp. (11%), 1 *Rhizobium* sp. (1.4%), 26 *Shewanella* spp. (36.1%), 2 *Stenotrophomonas* spp. (2.8%), and 29 *Vibrio* spp. (40.3%) (Table 2). Fifty-three of the identified isolates were tested for their susceptibility to seven different antimicrobial agents (Table 3). Among 23 *Vibrio* spp. isolates, 22 (95.7%) were resistant to ampicillin. This included one isolate (*V. alginolylticus*) resistant to ampicillin and gentamicin and two isolates (8.7%) that displayed multidrug-resistant (MDR) phenotypes: *V. alginolyticus* (cefotaxime (CTX), ampicillin (AMP), gentamicin (GEN), and cefoxitin (FOX)) and *V. fluvialis* (CTX, AMP, amoxicillin–clavulanic acid (AMC), FOX). Among the 26 *Shewanella* spp. isolates, none were resistant to any of the drugs tested, except for three isolates that had intermediate resistance to imipenem. Due to a lack of Clinical and Library Standards Institute (CLSI) interpretive guidelines for the disc-diffusion test, we were unable to test *Pseudomonas* spp. isolates for phenotypic resistance.

Of the 174 isolates that grew on drug-supplemented MacConkey agar plates, 174, 37, 98, and 11 isolates were tested for the presence of genes that encode class 1 integrons, carbapenemase (KPC), ESBLs (TEM, OXA, SHV), and CTX-M-type ESBLs, respectively. All PCR reactions were negative for these resistance genes.

## 3. Discussion

From 40 near-shore sites in the Bay Area, we isolated 18 distinct species of Gram-negative saprophytic bacteria (Table 2) on drug-supplemented plates. Although no recognized pathogenic GNB species were identified, many culturable isolates exhibited resistance to clinically used antimicrobial agents. Most studies assessing the presence of drug resistance in environmental bacteria thoroughly characterize a small number of sites, typically near known point-source pollutant effluent locations [1,17,18,26,34,35]. There have also been some comparative surveys across multiple water bodies [6,15]. In contrast to these designs, our sampling scheme extensively sampled a near-shore environment under ambient urban influence. In particular, our sampling sites were probabilistically chosen from intertidal and shallow subtidal areas around a large, urbanized estuary [36]. In this regionally representative sampling program, environmental bacteria were successfully isolated from every sampling site.

While species of several genera identified here (e.g., *Aeromonas* spp., *Pseudomonas* spp., *Shewanella* spp., and *Vibrio* spp.) have been described as opportunistic pathogens, they are all commonly found in marine-sediment environments, and their presence is rarely considered a public health risk [37,38,39,40,41]. Nevertheless, the genera *Aeromonas*, *Pseudomonas*, and *Shewanella* have been implicated as natural progenitors of, and reservoirs for, resistance genes such as CTX-M-, GES-, VIM-, and OXA-type ESBLs and carbapenemases that can be horizontally transferred into more pathogenic bacteria [25,42,43,44]. High rates of ampicillin resistance in *Vibrio* spp. have been well documented [45], consistent with the resistance rate of 96% found in this study. We also found five *Vibrio* isolates (22%) that displayed other resistance phenotypes. However, none of these harbored any of the common clinical resistance genes we tested for, including TEM, SHV, and OXA. The majority of bacteria that grew under selective pressure for imipenem resistance were *Shewanella* spp. (26 isolates; 74%); however, only three of these *Shewanella* spp. isolates (12%) exhibited intermediate resistance to carbapenems, with none being resistant. *Shewanella* spp. have been reported elsewhere to have reduced susceptibility to carbapenems, and the genus has also been identified as a natural progenitor of several OXA-type carbapenemases [42,46], yet none were found in our study.

Due to an absence of CLSI guidelines for nonclinically relevant bacteria, the antimicrobial susceptibilities of the isolated *Pseudomonas* spp. (including *Pseudomonas fluorescens*, *P. oleovorans*, *P. putida*, and *P. stutzeri*) were not tested by the disc-diffusion method. However, further investigation into the antimicrobial resistance profiles and genes for these isolates is warranted because the genus has been observed to harbor genes that mediate resistance to antimicrobial agents. Environmentally occurring *Pseudomonas* spp. harboring carbapenemases and ESBLs, namely VIM, IMP, and several CTX-M variants, have been widely reported in the past decade [19,43,47,48]. The CTX-M variants that we tested for in isolated *Pseudomonas* spp. were those known to be circulating in the region and were previously found in *P. putida* and *P. teessidea* in retail spinach [16].

A notable result of this study was the absence of drug-resistant bacteria from the Enterobacteriaceae family as well as the absence of fecal indicator bacteria [19]. A number of similar studies found an abundance of such bacteria, but these studies were conducted in water bodies and under conditions that would suggest a priori high levels of fecal contamination [4,17,40]. The frequency of fecal contamination in San Francisco Bay beaches is variable but generally low, and Bay beaches are typically safe for human recreation, with most beaches considered safe for swimming, especially during dry weather [32,33]. Carbapenem-resistant Enterobacteriaceae (CRE) present serious public health risk, and they were a major target of the present study; however, no such bacteria were isolated from the areas we tested.

There were several limitations in this study. Certain species may have been inhibited by the stress of the freeze–thaw step in combination with drug-supplemented MacConkey agar. Further, the techniques described here were culture-dependent, and PCR analysis was restricted to class 1 integrons and β-lactam resistance, which precluded the identification of other integrons or potentially relevant resistance mechanisms. Nevertheless, given the clinical importance of class 1 integrons [49], the observation of other associated resistance genes such as trimethoprim–sulfa or aminoglycoside would be unlikely in their absence. That said, the observed absence of β-lactam resistance mechanisms does not consider the full range of possible resistance genes. In the future, metagenomic study of DNA present in San Francisco Bay sediment samples or other whole-resistome screening approaches could reveal other clinically or environmentally relevant mechanisms [6,35].

Importantly, this study probabilistically sampled from 37 of the sites [36] in order to assess for regional patterns, rather than focusing only on areas of anthropogenic contamination. Studies that target wastewater treatment plants, hospital effluents, or animal livestock runoff could yield a higher prevalence of antimicrobial resistance among clinically relevant bacteria. In our study, the absence of clinically relevant drug-resistant GNB harboring β-lactamases and related resistance determinants suggests that GNB from ambient sediments in this well-managed, urbanized estuary may not constitute a major human exposure hazard at this time. These findings may be related to secondary and tertiary treatment operations and control measures for all wastewaters that drain into the Bay [28], in combination with the large dilution factor due to tidal exchange, resulting in low ambient sediment bacterial pollution in this estuary. These hypotheses could be tested in the future by evaluating resistance profiles and mechanisms in bacteria obtained from point sources and adjacent locations, including wastewater discharge effluents [1,4,18,34,35]. However, our study represents just one line of evidence, and routine water monitoring does periodically detect elevated fecal coliforms at some beaches [32]. Resistance to β-lactams continues to spread globally in GNB while, in parallel, novel resistance genes in environmental bacteria continue to be described. Therefore, routine environmental surveillance is needed to assess for the presence of potentially harmful bacteria or drug-resistance genes.

## 4. Materials and Methods

### 4.1. Sample Collection and Processing

Thirty-seven near-shore sites were sampled from Central San Francisco Bay (Central Bay), and three from Suisun Bay, both sub-basins of San Francisco Bay (Figure 1). The Central Bay sites were selected using a generalized random-tessellation stratified methodology, which is a probabilistic but spatially balanced method developed to identify locations for the sampling of natural resources [50]. The Suisun Bay samples were convenience samples, employing collection methods identical to those of the Central Bay sites. Although all sites were near-shore, a variety of habitats were included in the spatial sample, including both open water and narrow channels, sites adjacent to densely populated areas (e.g., San Francisco, CA, USA; Oakland, CA, USA), and sites proximate to more sparsely populated areas (e.g., Marin County and Suisun Bay) (Figure 1) [36].

Coastal Conservation and Research (Moss Landing, CA, USA) sampled all sites between 27 July and 14 September, 2015, as part of the Regional Monitoring Program (RMP)’s Bay Margins Sediment Study [51,52]. Sediments were collected by boat, with personnel using a modified VanVeen grab (0.1 m^2^ sampling area), from which 15 mL of surface sediment was scraped into a 50-mL conical tube (Fischer Scientific, Hampton, NH, USA). Sediment samples were combined with 20 mL of a preservative solution (15% glycerol in phosphate-buffered saline solution, PBS) and stored on dry ice (−78.5 °C) for transportation to the laboratory, after which they were immediately stored at −20 °C until analysis.

### 4.2. Gram-Negative Bacteria Isolation

Samples were thawed prior to analysis and diluted 10-fold with PBS. We selected for bacteria with reduced drug susceptibility by incubating 100 µL of this PBS–sediment solution on MacConkey agar (Difco Laboratories Inc., Detroit, MI, USA), supplemented with one of four antibiotics: ampicillin (16 µg mL^−1^), gentamicin (10 µg mL^−1^), imipenem (1 µg mL^−1^), or cefotaxime (1 µg mL^−1^). An additional plate without any antibiotics was used to assess baseline growth. Plates were incubated at 37 °C for 24 h and assessed for growth. In the absence of any growth at this stage, plates were incubated for another 24 h. All plates were examined for well-formed colonies, and the total number of CFUs was recorded (Table 1). Up to four colonies were selected from each antibiotic plate for further analysis. In an attempt to increase the diversity of species isolated, we tried to choose morphologically distinct isolates within a plate, based on visual observation. The selected colonies were streaked for isolation on MacConkey agar and incubated again at 37 °C for 24 h. An isolated colony from each of these plates was then streaked for isolation on Luria–Bertani (LB) agar (Difco Laboratories Inc., Detroit, MI, USA) and incubated at 37 °C for 24 h. Finally, an isolated colony from each LB plate was subcultured in 4 mL of LB broth (Difco Laboratories Inc., Detroit, MI, USA) at 37 °C for 24 h. A 1-mL aliquot of this culture was saved in a 15% glycerol stock, and a separate 1-mL aliquot was used to extract DNA: Bacteria were concentrated by centrifugation (60 s, 14,000 RPM), resuspended in water, and placed in a boiling water bath for 10 min; excess cell debris was collected by centrifugation (30 s, 14,000 RPM); and the supernatant containing the DNA was pipetted to a separate tube and stored at −20 °C before analysis.

### 4.3. Bacterial Species Identification and Drug-Susceptibility Tests

Bacteria were identified by 16S rRNA sequencing. PCR was first carried out with the primers 16s8F/16s806R18 (94 °C for 5 min, then 30 cycles of 94 °C for 30 s, 62 °C for 30 s, and 72 °C for 90 s) as previously described by Raphael et al. [16]. The amplified DNA products (approximately 800 bp) were sequenced on an Applied Biosystems 3730 DNA analyzer (Applied Biosystems, Foster City, CA, USA) at the University of California, Berkeley, DNA Sequencing Facility. Antimicrobial susceptibility profiles were assessed with a disc-diffusion assay according to the CLSI interpretive guidelines [53].

All isolates were tested for the presence of genes encoding class 1 integrons by PCR following a procedure previously described by Raphael et al. [16]. Class 1 integrons were chosen because of their clinical relevance and prevalence and widespread distribution in Gram-negative bacteria, both globally [49] and in the San Francisco Bay region [10]. All isolates obtained from MacConkey agar plates containing ampicillin (16 μg mL^−1^), and those that were resistant to ampicillin by the disc-diffusion test, were examined by PCR for the presence of the following extended spectrum β-lactamase variants: TEM (including TEM-1 and TEM-2), SHV (including SHV-1), and OXA (OXA-1, OXA-4, and OXA-30). For this, we employed multiplex primers and reaction conditions described by Dallenne et al. [54]. Isolates obtained from MacConkey agar plates containing cefotaxime (1 μg mL^−1^), and those that were resistant to cefotaxime by the disc-diffusion test, were tested for CTX-M genes using a set of multiplex primers and conditions for CTX-M variants (CTX-M-1, CTX-M-3, and CTX-M-15) as described by Adams-Sapper et al. [22]. Isolates obtained from plates supplemented with imipenem were tested for the variants of the carbapenemase gene KPC using primers and conditions described by Dallenne et al. [54]. Additionally, all bacteria with multidrug-resistant phenotypes were tested for variants of all the above-mentioned genes: TEM, SHV, OXA, CTX-M, and KPC.

## Figures and Tables

**Figure 1 antibiotics-09-00400-f001:**
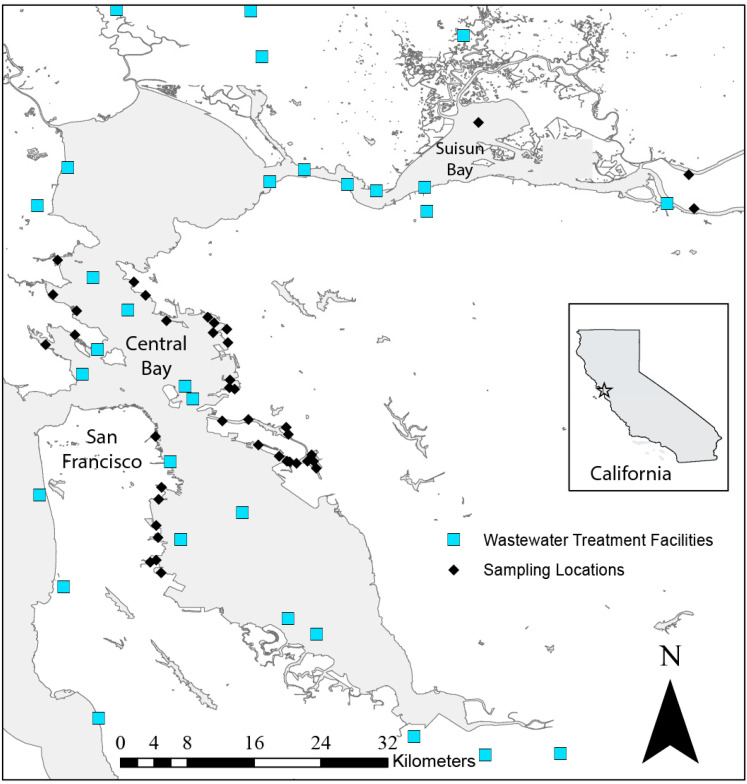
San Francisco Bay. Black diamonds (♦) indicate sediment collection location. Light blue squares (■) indicate wastewater treatment discharge locations in the region. Inset: Location within California, USA.

**Table 1 antibiotics-09-00400-t001:** Number of morphologically distinct bacterial colonies isolated from estuarine sediments in the San Francisco Bay Area, 2015, by antibiotic used for screening.

Antibiotic ^a^	Sites with Growth (N, %)	CFU/g ^b^	Sites with Isolates Obtained (N)	Morphologically Distinct Isolates Obtained (N) ^c^
No antimicrobial agent	40 (100%)	3513		
Ampicillin	34 (85%)	1280	32	87
Cefotaxime	9 (22.5%)	16	8	11
Imipenem	15 (37.5%)	106	15	37
Gentamicin	27 (67.5%)	196	13	39

^a^ Concentrations of antibiotic embedded in MacConkey agar plates: ampicillin, 16 µg mL^−1^; imipenem, 1 µg mL^−1^; cefotaxime, 1 µg mL^−1^; and gentamicin, 10 µg mL^−1^. ^b^ Colonies were counted on MacConkey agar plates and multiplied by the dilution factor to approximate the number of CFU/g sediment in each sediment sample. ^c^ Number of bacteria isolated from all antibiotic screening plates.

**Table 2 antibiotics-09-00400-t002:** Identity of bacterial species recovered from San Francisco Bay sediment, 2015, by antibiotic used to select for resistance in initial MacConkey agar plate.

Antibiotic	Species	Isolates (N)	Antibiotic	Species	Isolates (N)
Ampicillin (16 µg mL^−1^)	Total	21	Imipenem (1 µg mL^−1^)	Total	35
	*Vibrio alginolyticus*	6		*Aeromonas australiensis*	1
	*Vibrio parahaemolyticus*	6		*Aeromonas hydrophila*	1
	*Vibrio alginolyticus/parahaemolyticus* ^a^	7		*Aeromonas veronii*	1
	*Vibrio alginolyticus/azureus* ^a^	2		*Castellaniella defragrans*	1
Cefotaxime (1 µg mL^−1^)	Total	11		*Pseudomonas* sp. ^b^	1
	*Acinetobacter venetianus*	1		*Shewanella algae*	7
	*Gallaecimonas xiamenensis*	1		*Shewanella algae/haliotis* ^a^	11
	*Pseudomonas fluorescens*	2		*Shewanella loihica*	8
	*Pseudomonas oleovorans*	1		*Stenotrophomonas maltophilia*	2
	*Pseudomonas putida*	3		*Vibrio diazotrophicus*	1
	*Pseudomonas stutzeri*	1		*Vibrio fluvialis*	1
	*Rhizobium* sp. ^b^	1	Gentamicin (10 µg mL^−1^)	*Vibrio parahaemolyticus*	5
	*Vibrio fluvialis*	1			

^a^ Unable to discriminate between two species after 16S sequence analysis. ^b^ Species not determined.

**Table 3 antibiotics-09-00400-t003:** Species and antibiotic resistance profiles of bacteria recovered from estuarine sediments in San Francisco Bay, 2015, from drug-supplemented media. Plate: Drug supplementation on plate (see Methods).

Species	Isolates (N)	Plate	Resistance (Disc Diffusion) ^a^	Intermediate Resistance (Disc Diffusion) ^a^
*Acinetobacter venetianus*	1	CTX	CTX	
*Aeromonas australiensis*	1	IPM	AMC	
*Aeromonas hydrophila*	1	IPM	FOX	AMC
*Aeromonas veronii*	1	IPM	None	
*Shewanella algae*	4	IPM	None	
*Shewanella algae*	3	IPM		IPM
*Shewanella algae/halitosis* ^b^	11	IPM	None	
*Shewanella loihica*	8	IPM	None	
*Vibrio alginolyticus*	3	AMP	AMP	
*Vibrio alginolyticus*	1	AMP	AMP, CTX, GEN, FOX	
*Vibrio alginolyticus*	1	AMP	AMP, GEN	
*Vibrio alginolyticus/parahaemolyticus* ^b^	4	AMP	AMP	
*Vibrio diazotrophicus*	1	IPM	None	
*Vibrio fluvialis*	1	CTX	AMP, CTX, AMC	FOX
*Vibrio fluvialis*	1	IPM	AMP	AMC
*Vibrio parahaemolyticus*	5	AMP	AMP	
*Vibrio parahaemolyticus*	1	AMP	AMP	CIP
*Vibrio parahaemolyticus*	5	GEN	AMP	

^a^ AMP, ampicillin; CTX, cefotaxime; IPM, imipenem; AMC, amoxicillin–clavulanic acid; GEN, gentamicin; FOX, cefoxitin; CIP, ciprofloxacin. ^b^ Unable to discriminate between two species after 16S sequence analysis.

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
