# Peer review of "Antimicrobial Drug-Resistant Gram-Negative Saprophytic Bacteria Isolated from Ambient, Near-Shore Sediments of an Urbanized Estuary: Absence of β-Lactamase Drug-Resistance Genes"

_antibiotics, 2020, doi:10.3390/antibiotics9070400_

Round 1

Reviewer 1 Report

This is a concise and interesting report that should be general interest to the readers of "Antibiotics". Moritz et al. cultured Gram-negative bacteria from sediments from 40 locations within San Francisco Bay, notably from sites distant from wastewater treatment facilities. Cultures were selected for resistance to one of four different antibiotics, and the strains isolated were identified by 16S rRNA sequencing, and cross-resistance to seven antibiotics examined. None of the species were recognized pathogens. Finally, they were tested for the presence of class 1 integrons and several broad spectrum beta-lactam resistance genes. None of these resistance determinants was found.

This is a well organized, clearly written manuscript, and the findings are discussed in comparison with other similar studies from higher-risk environmental locations. I have no suggestions for improvement.

Author Response

We thank the reviewer for taking the time to review the manuscript, and for their kind words, and positive and thoughtful overall assessment of the work. No changes have been requested by this reviewer.

Reviewer 2 Report

Brief summary

Moritz and colleagues conducted a probabilistic survey in environmental samples collected from 40 intertidal and shallow subtidal areas around San Francisco Bay, in order to assess the prevalence of antimicrobial resistance and screen for clinically relevant β-lactamase resistance determinants in Gram-negative bacteria. The authors retrieved 74 isolates, and tested 55 of them for antimicrobial resistance; the Vibrio spp. showed the most notable resistance profiles. The authors did not identify clinically relevant drug resistant Gram-negative bacteria or mobile β-lactam resistance determinants.

Broad comments

This is a clearly written and well presented manuscript with only a minor editing flaw: the methods are presented after the discussion, while they should appear before results. The abstract is informative and clear, and the other sections of the main manuscript are well organized and comprehensive. Data provided in this manuscript are well summarized. The tables and the figure are comprehensive and helpful. Overall, it is a well written and insightful manuscript.

A comparative study with collection sites that are adjacent to treated wastewater discharge locations, or collection from areas without reliable secondary or tertiary wastewater treatment facilities would have provided informative data on how wastewater facilities could prevent (or not) the pollution and the development of resistant microorganisms in coastal and river waters located in populated areas. This could constitute a future research project and should not (by any means) preclude the publication of this manuscript.

Author Response

We thank the reviewer for taking the time to review the manuscript, for their positive evaluation of the manuscript, and for providing thoughtful and constructive comments.

Review Comment 2a:  “This is a clearly written and well presented manuscript with only a minor editing flaw: the methods are presented after the discussion, while they should appear before results.”

Response: The journal formatting template specifies that the Methods section should be presented after Discussion section, as done here. While the proposed change is certainly quite reasonable, we want to follow the standard format for this journal. No change made.

Review Comment 2b:  “A comparative study with collection sites that are adjacent to treated wastewater discharge locations, or collection from areas without reliable secondary or tertiary wastewater treatment facilities would have provided informative data on how wastewater facilities could prevent (or not) the pollution and the development of resistant microorganisms in coastal and river waters located in populated areas. This could constitute a future research project and should not (by any means) preclude the publication of this manuscript.”

Response: We agree that this would be a useful study in the future. As such, we add the following sentence along with appropriate references to the last paragraph of the Discussion (line 183): “These hypotheses could be tested in the future by evaluating resistance profiles and mechanisms in bacteria obtained from point sources and adjacent locations, including wastewater discharge effluents.”

Reviewer 3 Report

In the paper the authors found that that Gram-negative bacteria from ambient sediments in this well-managed, urbanized estuary are unlikely to contain either clinically relevant strains or β-lactam resistance gene variants.

I find this study interesting and deserving publication although I have some comments for the consideration of the authors.

Abstract section: rewrite the conclusion are intuitive and not original

Introduction section: add the reference. For example the following reference could be add:

Vasant Nagvekar et al. Prevalence of multi drug resistant Gram-negative bacteria cases at admission in multispecialty hospital. Journal of Global Antimicrobial Resistance. Available online 9 March 2020.

Fasciana et al. Co-existence of virulence factors and antibiotic resistance in new Klebsiella pneumoniae clones emerging in south of Italy BMC Infectious Diseases Volume 19, Issue 1, 4 November 2019.

Materials and methods. Add the criteria for strains isolation

Add the  conclusions section

Author Response

We thank the reviewer for taking the time to review the manuscript and for providing their constructive review comments. We feel that these comments and our consequent revisions (described below) have improved the manuscript.

Review Comment 3a: “Abstract section: rewrite the conclusion are intuitive and not original”

Response: We agree that the abstract could end with a more original statement. The last sentences of the abstract in the original submission previously read, “Thus, while drug resistant bacteria were recovered in our study, neither clinically relevant drug resistant Gram-negative bacteria nor mobile β-lactam resistance determinants were found. Our findings indicate that Gram-negative bacteria from ambient sediments in this well-managed, urbanized estuary are unlikely to contain either clinically relevant strains or β-lactam resistance gene variants.” We have revised these sentences (lines 29 – 33) to reduce redundancy and increase novelty as follows, “Thus, while drug resistant Gram-negative bacteria were recovered from ambient sediments, neither clinically relevant strains nor mobile β-lactam resistance determinants were found. This suggests that Gram-negative bacteria in this well-managed, urbanized estuary are unlikely to constitute a major human exposure hazard at this time.”

Review Comment 3b: “Introduction section: add the reference. For example the following reference could be add:

Vasant Nagvekar et al. Prevalence of multi drug resistant Gram-negative bacteria cases at admission in multispecialty hospital. Journal of Global Antimicrobial Resistance. Available online 9 March 2020.

Fasciana et al. Co-existence of virulence factors and antibiotic resistance in new Klebsiella pneumoniae clones emerging in south of Italy BMC Infectious Diseases Volume 19, Issue 1, 4 November 2019.”

Response: We have added the two aforementioned references and have also added an additional reference to broaden the review of global spread of resistance in the Introduction (first paragraph, line 42).

Review Comment 3c: “Materials and methods. Add the criteria for strains isolation”

Response: We agree that this information is warranted. We have therefore modified and expanded the text from original submission. The revised text reads as follows: “All plates were examined for well-formed colonies and the total number of CFUs was recorded (Table 1). Up to four colonies were selected from each antibiotic plate for further analysis. In an attempt to increase the diversity of species isolated, we tried to choose morphologically-distinct isolates within a plate based on visual observation. The selected colonies were streaked for isolation on MacConkey agar and incubated again at 37 °C for 24h. An isolated colony from each of these plates was then streaked for isolation on Luria-Bertani (LB) agar (Difco Laboratories Inc., Detroit, MI) and incubated at 37 °C for 24h. Finally, an isolated colony from each LB plate was subcultured in 4 ml of LB broth (Difco Laboratories Inc., Detroit, MI) at 37 °C for 24h.” The revised text is found in Section 4.2, lines 216 – 223 

Review Comment 3d: “Add the conclusions section”

Response: The suggestion is certainly quite reasonable but was not raised by the other two reviewers. In the formatting template provided to authors, the following statement was made regarding a conclusions section: “This section is not mandatory, but can be added to the manuscript if the discussion is unusually long or complex.” We do not feel that the present discussion (6 paragraphs; 914 words) requires a separate conclusion. In order to maintain a reasonably concise paper, we opt to maintain the current organization (Results followed by Discussion).